# LOCA: LOGICAL CHAIN AUGMENTATION FOR SCIENTIFIC CORPUS CLEANING

## ABSTRACT

While Large Language Models (LLMs) excel in general domains, their reliability often falls short in scientific problem-solving. The advancement of scientific AI depends on large-scale, high-quality corpora. However, existing scientific question-answering (QA) datasets suffer from high error rates, frequently resulting from logical leaps and implicit reasoning within the answers. To address this issue, we introduce LOCA (Logical Chain Augmentation), a novel framework for automatically cleaning scientific corpora, implemented through an augment-and-review loop. At its core, LOCA enhances raw answers by completing missing logical steps and explicitly separating the underlying scientific principle from its subsequent derivation. By applying LOCA to challenging scientific corpora, we demonstrate that it can automatically filter noisy datasets, typically reducing the error rate from as high as 20% to below 2%. LOCA provides a scalable and effective methodology for creating high-quality scientific corpora, paving the way for more reliable training and evaluation of scientific AI.

## 1 INTRODUCTION

LLMs have demonstrated remarkable capabilities across a wide range of general tasks(Brown et al., 2020; OpenAI, 2022; Achiam et al., 2023; Anil et al., 2023; Touvron et al., 2023a;b; Liu et al., 2024; Guo et al., 2025a; claude, 2025; Comanici et al., 2025; OpenAI, 2025a;b; Team, 2025; Team et al., 2025; Yang et al., 2025). However, their reliability often drops in domains demanding extreme accuracy, such as scientific problem-solving. The grand challenge of building artificial general intelligence capable of scientific discovery depends critically on the availability of high-quality, large-scale scientific corpora(Zheng et al., 2023). These corpora, typically structured as QA pairs, are the foundation for training more capable models and for creating reliable benchmarks to evaluate their progress(Huang et al., 2024).

Scientific problems, particularly in fields like physics, present unique difficulties. Unlike well-structured domains like mathematics or code, they require modeling complex real-world scenarios and unstructured reasoning, where the validity of a step depends heavily on context-sensitive scientific principles (Qiu et al., 2025; Feng et al., 2025; Zhang et al., 2025b; Siddique et al., 2025; Zhang et al., 2025a). This complexity leads to the frequent occurrence of errors in existing scientific corpora. For example, our own expert analysis reveals that error rates in major benchmarks' QA pairs can exceed 20% (Qiu et al., 2025; Feng et al., 2025).

We identify a primary reason for this unreliability: the logical incompleteness of provided solutions. Experts and AIs alike tend to omit steps they consider "obvious", inadvertently hiding subtle but critical flaws in their reasoning. While manual expert review offers a gold standard in principle, it is prohibitively slow and expensive, making it unable to scale. While some automated cleaning pipelines exist, they often focus on surface-level answer inconsistency (Wang et al., 2022b; Singh et al., 2023; Chen et al., 2024; Guo et al., 2025b; Hao et al., 2025; Riaz et al., 2025), failing to address the core issue in scientific reasoning: structural soundness and completeness of the logical argument itself.

To address this issue, we introduce LOCA, a novel framework for scientific corpus cleaning. The key insight is to enforce a detailed, structured and verifiable reasoning process by interpolating missing logical steps and decomposing each step into two orthogonal components: the underlying principle (e.g., a physical law) and its subsequent derivation (e.g., solving an equation). This structured augmentation not only improves the inherent clarity and correctness of the solutions but also enables a highly reliable review loop to flag suspicious QA pairs, and thus resulting in a high-accuracy cleaned corpus. Those rejected pairs, now with enhanced structure and readability, become significantly efficient to human expert review and correction.

We evaluate LOCA's performance on challenging physics QA pairs drawn from some existing high-quality corpora. Our experiments show that LOCA significantly reduces the residual error rate of the resulting cleaned corpus (< 2%), outperforming various baselines.

The contributions of our work can be summarized as follows:

- We propose LOCA, a novel framework that cleans scientific corpora by enforcing logical completeness and decomposing reasoning steps into verifiable principles and derivations, within an augment-and-review loop.

- We show that LOCA significantly reduces the residual error rate of the resulting cleaned corpus while retaining a large accepted set, outperforming various baselines.

- Our work offers a scalable path to building high-quality corpora for reliably training and evaluating scientific AI. Furthermore, LOCA's structured outputs can speed up expert content creation and serve as effective educational tools.

## 2 RELATED WORKS

**Corpus Cleaning and Augmentation.** What are commonly referred to as corpus cleaning approaches range from heuristic and model-based filtering (Soldaini et al., 2024; Penedo et al., 2024) to synthetic data generation via seed-based synthesis or corpus rephrasing (Abdin et al., 2024; Su et al., 2024). However, more directly related to our work are methods for reviewing and correcting flawed reasoning. These include self-correction pipelines where a model reflects on its own output (Madaan et al., 2023; Pan et al., 2025) and multi-agent debate frameworks where different agents critique solutions (Liang et al., 2023; Liu et al., 2025; Du et al., 2023). However, these approaches have known limitations: self-correction often misses subtle errors (Huang et al., 2023), and general debate frameworks can lack the domain-specific structure needed for complex scientific validation (Liu et al., 2025).

**Physics Corpora.** The inherent complexity of physics, characterized by its demand for multi-step reasoning and precise mathematical modeling, has motivated the development of specialized corpora, often released as benchmarks. These range from expert-curated, competition-level problems with detailed solutions in PHYBench (Qiu et al., 2025) to large-scale, university-level collections such as PHYSICS (Feng et al., 2025), ABench-Physics (Zhang et al., 2025b), PhysReason (Zhang et al., 2025a) and PhysicsEval (Siddique et al., 2025). These resources, though valuable for presenting challenging problems and striving for optimal answers, are critically limited by reference solutions that often contain logical leaps and errors. This fundamentally reduces their reliability for model training and evaluation, a gap our work aims to address.

## 3 METHOD

We propose LOCA, a framework that cleans and improves scientific corpora by converting the implicit reasoning in each answer into detailed, structured steps that provide a verifiable basis for accepting or rejecting the answer. The pipeline of LOCA is shown in Figure 1.

Figure 1: **Pipeline of LOCA.** LOCA employs an iterative augment-and-review loop. In each iteration, given a raw answer with some reasoning process ($A_{\text{aug}}$), an augmentation agent structures it through chain completion and structured decomposition; this structured output is then assessed by specialized review agents. Based on the feedback, the answer is either refined for the next iteration, accepted after passing multiple checks, or rejected. Accepted answers undergo a final external consistency check against $A_{\text{raw}}$, while rejected ones can be flexibly routed to human experts for review.

## 3.1 LOGICAL CHAIN AUGMENTATION

The cornerstone of LOCA is logical chain augmentation, which transforms a raw, unstructured answer with some reasoning process into a structured, verifiable logical chain. Formally, consider a question $Q$ and its raw answer, represented as a sequence of steps $A_{\text{raw}} = (s_1, \ldots, s_n)$, where each step $s_i$ represents an implicit transformation from a context $C_{i-1}$ to $C_i$:

$$s_i : C_{i-1} \to C_i. \tag{1}$$

Here, $C_n$ contains the final result. Typically, $A_{\text{raw}}$ suffers from two key limitations:

- **Non-Atomicity**. A single step $s_i$ often conflates multiple atomic reasoning steps into a "logical leap," which not only introduces a higher risk of errors but also hinders their precise localization.

- **Implicit Justification**. The rationale for a step (the "why," e.g., a physical law) is often entangled with its subsequent mathematical derivation (the "how", e.g., solving an equation) or omitted, obscuring the reasoning process.

To address these issues, LOCA employs a transformation implemented through an LLM-based agent, to map the raw answer $A_{\text{raw}}$ to an augmented answer $A_{\text{aug}}$. This transformation is achieved through two intertwined operations: **chain completion** and **structured decomposition**.

**Chain Completion.** This operation enforces atomicity by completing reasoning steps that are often omitted by both human experts and LLMs. LOCA addresses this by making all implicit assumptions, intermediate conclusions, and derivations explicit. Formally, a step $s_i : C_{i-1} \to C_i$ is non-atomic if it implicitly contains an intermediate context $C_{\text{int}}$ (i.e., $C_{i-1} \to C_{\text{int}} \to C_i$). LOCA identifies and decomposes each such step into a more fundamental subsequence $(s'_{i,1}, s'_{i,2}, \ldots, s'_{i,k})$ where $k > 1$:

$$\forall s_i \in A_{\text{raw}}, \text{ if } \neg \text{IsAtomic}(s_i), \text{ then } s_i \mapsto (s'_{i,1}, \ldots, s'_{i,k}). \tag{2}$$

This results in a new, more detailed sequence $S' = (s'_1, s'_2, \ldots, s'_m)$ with $m \geq n$, where each step represents a more atomic inference, significantly reducing the chance of hidden errors within a single step.

**Structured Decomposition.** To distinguish the underlying logical assumption from its specific application, we decompose each atomic step $s'_j$ into two orthogonal components: principle ($P_j$) and derivation ($D_j$). This design is inspired by formal proof systems where each inference applies a specific rule to a set of premises (Moura & Ullrich, 2021). Each step in the augmented answer $A_{\text{aug}}$ thus becomes a tuple:

$$s'_j = (P_j, D_j). \tag{3}$$

- **Principle** ($P_j$). A declarative statement of the step's logical foundation from an axiom space $\mathbb{P}$:

$$P_j \in \mathbb{P} = \{\text{Newton's Second Law, L'Hôpital's Rule, Geometric Constraints ...}\}. \tag{4}$$

  $P_j$ answers the question: "Why can this step be taken?"

- **Derivation** ($D_j$). The derivation required to apply principle $P_j$ to the preceding context $C_{j-1}$ and yield the new context $C_j$:

$$D_j = \mathcal{D}(C_{j-1}|P_j), \qquad C_j = C_{j-1} \cup P_j \cup D_j. \tag{5}$$

  $D_j$ answers the question: "How is this principle specifically applied?"

The final augmented answer is thus a sequence of these structured tuples:

$$A_{\text{aug}} = ((P_1, D_1), (P_2, D_2), \ldots, (P_m, D_m)). \tag{6}$$

## 3.2 Iterative Augmentation and Review

The structured feature of $A_{\text{aug}}$ provides a powerful foundation for an iterative cycle of augmentation and review. This loop is designed to progressively improve a solution until it meets a high standard of logical rigor or is definitively flagged as flawed. Notably, the primary goal of this process is not necessarily to increase the answer's correctness, but rather to enable a more accurate *judgment*. Along with the final decision criterion described later, LOCA ensures that the ultimately accepted QA pairs have an exceptionally low residual error rate while maintaining a substantial set size.

In each iteration, the correctness of each step $s'_j$ is reviewed by two specialized components—principle review ($\mathcal{R}_P$) and derivation review ($\mathcal{R}_D$):

$$\mathcal{R}(s'_j) = \mathcal{R}_P(P_j|C_{j-1}) \wedge \mathcal{R}_D(D_j|P_j, C_{j-1}). \tag{7}$$

- **Principle Review** ($\mathcal{R}_P$). This assesses the validity of introducing principle $P_j$ in the given context $C_{j-1}$ (e.g., checking if energy conservation is valid when non-conservative forces exist).

- **Derivation Review** ($\mathcal{R}_D$). This assesses the correctness of derivation $D_j$, assuming $P_j$ is valid. This is a symbolic or mathematical check ensuring the principle was applied accurately.

In practice, we instantiate $\mathcal{R}_P$ and $\mathcal{R}_D$ using specialized LLM agents respectively. This decomposition is crucial as it isolates distinct sources of error, thereby enabling more sensitive error detection. The integrated $\mathcal{R}$ not only yields a binary judgment (correct/wrong) but also provides detailed feedback for further augmentation.

Leveraging this verifiable structure, LOCA employs an iterative augment-and-review loop. The loop terminates once its status is finalized under one of two terminal conditions:

- **Passed**. $A_{\text{aug}}$ passes review when $N_{\text{corr}} \geq N_{\text{corr}}^{(\text{max})}$, where $N_{\text{corr}}$ counts *consecutive* iterations satisfying $\bigwedge_j \mathcal{R}(s'_j)$, and $N_{\text{corr}}^{(\text{max})}$ is a hyperparameter.

- **Failed**. $A_{\text{aug}}$ fails review when $N_{\text{wrg}} \geq N_{\text{wrg}}^{(\text{max})}$, where $N_{\text{wrg}}$ is the *cumulative* count of iterations not satisfying $\bigwedge_j \mathcal{R}(s'_j)$, and $N_{\text{wrg}}^{(\text{max})}$ is a hyperparameter.

If $\bigwedge_j \mathcal{R}(s'_j)$ is not satisfied and $N_{\text{wrg}} < N_{\text{wrg}}^{(\text{max})}$ in an iteration, LOCA resets $N_{\text{corr}} = 0$ and then leverages the feedback to refine the augmented solution for the next iteration. This mechanism prevents the premature acceptance of flawed answers while using targeted feedback for improvement.

## 3.3 Final Decision Criterion for Corpus Partitioning

The LOCA pipeline partitions the initial corpus into two disjoint sets: **accepted** and **rejected**. A QA pair is accepted if and only if it satisfies two criteria:

- **Internal Coherence**. The iterative review process must terminate as the "passed" state. This confirms the augmented solution is logically stable and self-consistent.
- **External Consistency**. The final result of the augmented answer ($A_{\text{LOCA}}$) must match that of the original answer ($A_{\text{raw}}$), i.e., $A_{\text{LOCA}} \equiv A_{\text{raw}}$. Any mismatch indicates a potential error in either the original or the augmented answer, flagging the QA pair as temporarily unreliable.

Notably, the rejected set, containing structured answers $A_{\text{aug}}$, also provides a valuable resource for efficient human expert review and can be added back to the corpus upon correction.

## 4 Experiments

To evaluate LOCA's performance, we apply it to a diverse set of physics QA corpora—a representative case in natural sciences—and perform a comprehensive comparison against several mainstream reasoning and review methods. We mainly evaluate each method's capacity to filter a corpus by measuring the residual error rate within their respective accepted sets.

## 4.1 Baselines

The two components of our final decision criterion—external consistency and internal coherence—naturally align with distinct classes of mainstream reasoning and review methods. This correspondence provides a clear basis for structuring our comparison, and we therefore group our baselines into three categories.

**Reasoning-Based.** This category focuses exclusively on the external consistency. For a given problem, these methods generate a new solution from scratch, ignoring the raw answer. A QA pair is accepted only if the newly generated answer, $A_{\text{new}}$, matches the original answer, $A_{\text{raw}}$.

To establish baselines, we generate $A_{\text{new}}$ by employing several mainstream powerful reasoning strategies:

- **Direct Prompting.** The LLM is prompted to solve the problem in a single pass.
- **Chain-of-Thought (CoT).** We evaluate both Zero-Shot-CoT (Kojima et al., 2022), which appends "Let's think step by step" to the prompt, and Few-Shot CoT (Wei et al., 2022), which provides in-context examples of step-by-step reasoning.
- **Self-Consistency (CoT-SC).** An extension of CoT where multiple reasoning paths are sampled. The final answer $A_{\text{new}}$ is determined by a majority vote over the outcomes of these paths (Wang et al., 2022a). We sample $k = 5$ paths for this baseline.
- **Tree-of-Thoughts (ToT).** This method moves beyond linear reasoning by exploring a tree of intermediate thoughts. It allows the model to evaluate different reasoning steps (Yao et al., 2023). We configure ToT with a tree depth of $d = 4$ and a node size limit of $k = 2$.
- **Graph-of-Thoughts (GoT).** An extension of ToT that models the reasoning process as a more flexible graph structure, allowing for the merging of different reasoning paths (Besta et al., 2024).
- **Multi-Agent Debate (MAD).** This involves multiple LLM agents that propose and critique solutions in a structured debate format (Liang et al., 2023). We use 2 agents debating for 3 rounds.

**Review-Based.** This method isolates the condition of internal coherence in a simplified form. An LLM directly reviews the raw answer to judge its correctness. To mitigate the unreliability of a single review, we adopt a self-consistency approach. A QA pair is accepted only if the LLM judges the answer correct in $N_{\text{corr}}^{(\text{max})} = 3$ consecutive, independent evaluations. We denote this method as Review-SC.

**Iterative Self-Reflection.** The third category represents methods that integrate both conditions. We adopt the most representative self-reflection method which employs a feedback-refine loop to improve upon the raw answer (Madaan et al., 2023). In fact, LOCA itself also belongs to this category and we set $N_{\text{corr}}^{(\text{max})} = 3$, $N_{\text{wrg}}^{(\text{max})} = 5$ for evaluation.

## 4.2 DATASETS

To evaluate the general applicability of LOCA, we test it on a diverse collection of physics QA pairs sourced from several recent, high-quality physics benchmarks.

- **PHYBench (Qiu et al., 2025).** This contains competition-level problems created and reviewed by a large group of competition-trained students, consisting of gold medal–level competitors. For our experiments, we utilize the publicly available set of **100** problems that include complete, human-authored detailed solutions.
- **PHYSICS (Feng et al., 2025).** This consists of problems from publicly available PhD-qualifying exams. We randomly sample **100** problems from its text-only questions, specifically excluding proofs tasks.
- **ABench-Physics (Zhang et al., 2025b).** This contains university- and competition-level physics problems. Since the original dataset provides only answers without intermediate reasoning, we randomly sample **100** questions from this set and prompt GPT-5 to produce the corresponding raw answers with reasoning, thereby creating the complete QA pairs for our evaluation. This simulates a realistic scenario of cleaning corpora with LLM-generated reasoning.

LOCA's structured, complete logical-chain outputs enable our human experts to efficiently identify and correct erroneous answers in the datasets. Using this method, we manually detected 20, 22, and 13 incorrect answers in the three benchmarks, respectively, and provided correct answers wherever possible. These results serve as the ground-truth for the subsequent evaluation.

Table 1: **Performance comparison on PHYBench**. We report the residual error rate (%) of the accepted set, with its size (number of QA pairs) in parentheses. An ideal method minimizes the error rate, our primary focus, while maximizing the size of accepted set. Bold indicates the best result for each LLM. LOCA with Gemini 2.5 Pro significantly reduces the error rate while retaining a large accepted set.

| Method | Gemini 2.5 Pro | o3 | DeepSeek-R1 | GPT-5 |
|---|---|---|---|---|
| Direct Prompting | 6.25% (48) | 10.00% (30) | 7.14% (42) | 12.24% (49) |
| Zero-Shot-CoT | 11.76% (51) | 12.50% (32) | 9.30% (43) | 15.91% (44) |
| Few-Shot CoT | 7.84% (51) | 13.16% (38) | **5.71%** (35) | 10.20% (49) |
| CoT-SC | 10.42% (48) | 12.12% (33) | 9.09% (33) | 12.90% (31) |
| ToT | 9.43% (53) | 16.00% (25) | 8.57% (35) | 8.16% (49) |
| GoT | 7.14% (42) | 14.81% (27) | 9.09% (22) | 13.04% (23) |
| MAD | 8.57% (70) | 9.62% (52) | 10.71% (56) | 7.02% (57) |
| Review-SC | 12.66% (79) | 15.38% (78) | 17.05% (88) | 14.63% (82) |
| Self-Reflection | 10.39% (77) | 12.50% (80) | 16.46% (79) | 11.84% (76) |
| LOCA (ours) | **1.69%** (59) | **4.26%** (47) | 10.26% (39) | **6.56%** (61) |

Table 2: **Comprehensive comparison across various datasets.** Gemini 2.5 Pro is used for all cases, and results are also presented as: residual error rate (%) (accepted set size). Bold indicates the best method for each dataset. LOCA consistently achieves the lowest error rates across three datasets, demonstrating robust performance.

| Method | PHYBench | PHYSICS | ABench-Physics |
|---|---|---|---|
| Direct Prompting | 6.25% (48) | 9.52% (63) | 10.26% (78) |
| Zero-Shot-CoT | 11.76% (51) | 10.17% (59) | 10.00% (80) |
| Few-Shot CoT | 7.84% (51) | 6.67% (60) | 11.69% (77) |
| CoT-SC | 10.42% (48) | 8.20% (61) | 6.67% (75) |
| ToT | 9.43% (53) | 6.90% (58) | 7.50% (80) |
| GoT | 7.14% (42) | 6.78% (59) | 6.67% (75) |
| MAD | 8.57% (70) | 6.25% (48) | 8.70% (69) |
| Review-SC | 12.66% (79) | 12.35% (81) | 8.51% (94) |
| Self-Reflection | 10.39% (77) | 11.11% (72) | 7.61% (92) |
| LOCA (ours) | **1.69%** (59) | **1.64%** (61) | **1.22%** (82) |

## 4.3 RESULTS

**Preliminary Evaluation on PHYBench.**    In Table 1, a comparison across models reveals that Gemini 2.5 Pro generally provides the most favorable balance between achieving a low residual error rate and retaining a substantial number of QA pairs on various methods used. Besides, the combination of LOCA with Gemini 2.5 Pro achieves the lowest residual error rate among all combinations. Given the superior and robust performance of Gemini 2.5 Pro, we representatively select it for subsequent comprehensive experiments.

**Comprehensive Evaluation across Various Datasets.**    As shown in Table 2, LOCA substantially outperforms all baselines across three datasets. It significantly reduces the residual error rate to below 2% while retaining a substantial number of accepted QA pairs. The most telling comparison is against Self-Reflection (Madaan et al., 2023), which shares a similar framework with LOCA but lacks our logical chain augmentation. Self-Reflection yields a ∼10 times larger error rate. Besides, the performance of baselines also

Table 3: **Ablation study on LOCA's core components.** We evaluate variants by replacing the structured augmentation module, the specialized review module, or both. The results demonstrate that both components are critical and interdependent for minimizing the residual error rate.

| Method | PHYBench | PHYSICS | ABench-Physics |
|---|---|---|---|
| w/o Structured Augmentation | 10.14% (69) | 6.35% (63) | 4.82% (83) |
| w/o Specialized Review | 4.48% (67) | 9.59% (73) | 4.71% (85) |
| w/o Both | 4.62% (65) | 6.15% (65) | 6.59% (91) |
| LOCA (ours) | **1.69%** (59) | **1.64%** (61) | **1.22%** (82) |

underscores the necessity of LOCA's final decision criterion. Reasoning-based methods, relying solely on external consistency, exhibit high error rates (ranging from 6.25% to 11.76%). This approach can create a "false agreement", where $A_{new} \equiv A_{raw}$ not because both are correct, but because they might share a common mistake. Similarly, Review-SC, which isolates internal coherence, also proves unreliable (8.51% to 12.66%).

**Ablation Study.** To evaluate the individual contributions of LOCA's two core components—structured augmentation and specialized review—we conduct a comprehensive ablation study. In this study, we replace the structured augmentation and/or review module(s) with generic counterparts: a feedback-based refine module and a holistic review module. As shown in Table 3, replacing either component leads to a significant performance collapse. For instance, on PHYBench, the error rate increases from 1.69% to 10.14% without structured augmentation; on PHYSICS, replacing only the specialized review yields a 9.59% error rate, which is even worse than replacing both modules. This highlights a crucial point: the components are not merely additive but are tightly coupled. The observation that replacing just one part can be sometimes more harmful than replacing both underscores their strong interdependence. This validates that the co-design is essential to LOCA's effectiveness.

**Impact of LOCA's Cleaning on Model Capability Evaluation.** LOCA enables a more faithful and accurate evaluation of LLM performance. To demonstrate this, we evaluate several models on 3 versions of PHYBench, as shown in Figure 2: (1) the original Raw set; (2) a Filtered high-accuracy subset containing only QA pairs accepted by LOCA with Gemini 2.5 Pro; and (3) a Corrected version where flawed QA pairs have been fixed by human experts, powered by LOCA's augmentation. The results reveal two key findings. First, all models achieve their highest scores on the Filtered subset, which is expected as this set excludes potentially problematic or harder questions. More importantly, model scores on the Corrected benchmark are also substantially higher than on the Raw version. For instance, Gemini 2.5 Pro's score increases from 42.0 to 50.3, closer to the human baseline (61.9) reported (Qiu et al., 2025). This score improvement, derived from the same set of 100 questions, confirms that the original benchmark's errors are masking the model's true performance. These corrected scores provide a more holistic and accurate measure of a model's capabilities. Our findings underscore that a robust process like LOCA is critical not just for filtering, but for creating more reliable benchmarks.

## 5 CONCLUSION

In this work, we introduced LOCA, a novel framework designed to address the issue of high error rates in scientific QA corpora. By automatically enforcing logical completeness and decomposing complex reasoning into explicit principles and derivations, LOCA employs an augment-and-review loop to effectively filter noisy datasets while refining them for straightforward human expert verification. Our comprehensive experiments on challenging physics problems demonstrate that LOCA can reduce the residual error rate from

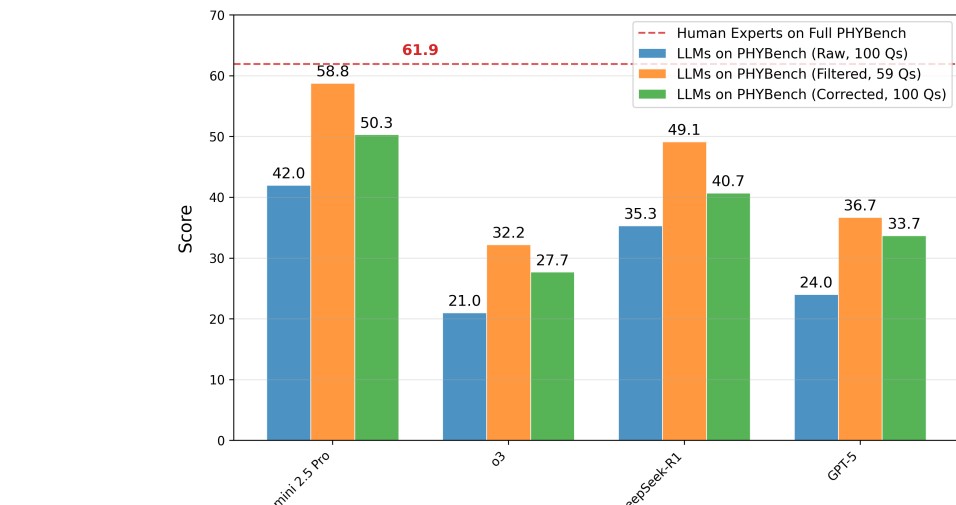

Figure 2: **Impact on evaluating LLMs' performance.** We compare model performance on 3 versions of PHYBench: the original Raw set (100 Qs); the high-accuracy Filtered subset accepted by LOCA (59 Qs); and the Corrected set (100 Qs), where flawed QA pairs are manually fixed powered by LOCA's augmentation.

typically 20% to below 2%, substantially outperforming a wide range of mainstream reasoning and review baselines. Furthermore, we show that benchmarks cleaned by LOCA enable a more accurate and faithful evaluation of LLM capabilities.

Although demonstrated in physics, LOCA's methodology is broadly applicable to other principle-based fields like chemistry and engineering. By providing a robust pathway to generating large-scale, high-quality scientific corpora, LOCA represents a key advance toward more trustworthy scientific AI. The structured, logically coherent outputs it produces also hold significant educational value.

**Limitations.** LOCA focuses on checking the correctness of *answers*. A limitation is that a small fraction of questions in scientific corpora can be ill-defined, ambiguous, or factually incorrect. Such flawed questions can interfere with the review process, as the notion of a "correct" solution becomes ambiguous.

**Reproducibility Statement.** The source code is available in the supplemental materials.

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

## A APPENDIX

### A.1 PROMPTS

We applied the LOCA workflow to a range of benchmark problems, employing a structured set of prompt templates to delegate distinct tasks to specialized agents. These tasks correspond to three primary roles: augmentation, review, and generation.

The prompt design for each role in the workflow is detailed below.

The proposed solution is first forwarded to the augmentation agent. Its responsibility is to enhance the solution by supplementing the reasoning chain and addressing any issues identified in the provided bug report. If no bug report is provided, the corresponding input field remains empty, and the agent proceeds without explicit error corrections.

---

### Augmentation Prompt

You are an AI expert specializing in physics problem-solving. Your
   task is to take a given physics problem, its potentially
   incomplete solution (Note that the original answer is not
   necessarily incomplete) and a report of bugs in this solution (
   Note that if the report shows no bugs, it is not necessarily no
    bugs), and produce a complete, step-by-step 'Refined Solution
   '. This refined solution MUST fix ALL bugs in the report, by
   adding or modifying missing steps, statements, principles,
   derivations, etc., to make the solution completed according to
   the problem statement and also make the logic clear, rigorous,
   and self-contained.
You should Not modify the final answer unless there is an
   inevitable contradiction during the derivation process.

You MUST adhere to the following strict formatting rules for your
   output:

1.  **Start Marker**: The entire output MUST begin *exactly* with
   the line `# Refined Solution` and nothing before it.
2.  **Sectioning**: The solution must be structured into sections
   using `###` headings.
3.  **First Section**: The first section MUST be titled `###
   Problem Statement Explanation`. In this section, you should
   detail and organize the information given in the problem
   statement, describe the physical situation / process, define
   all relevant variables one-by-one, and state any overall
   assumptions (e.g.,  approximations, geometric relations). You
   can not ignore any information in the problem statement.
4.  **Step Sections**: All following sections MUST be titled `###
   Step XXX`, where `XXX` is a number starting from 1 (e.g., `###
   Step 1`, `### Step 2`, ...). These sections should present the
   logical flow of the solution.
5.  **Content of Step Sections**: In the given solution, some
   parts are derivations or calculations, while others DIRECTLY
   adopt / introduce original formulas / assumptions / theorems /
   principles / geometric relations / boundary conditions. Within
   each `### Step XXX` section, you must follow this two-part
   structure:
-    **Principles/Original Formulas/Assumptions**: State all the
   original formulas / assumptions / theorems / physical

---

principles / geometric relations / approximations / boundary
conditions in their most original / general / universal /
standard forms(i.e., according to physical theories /
principles) being used in that step (maybe more than 1). Note
that this must be in their most original, general, universal,
or standard forms. Each of them MUST be placed on its own line
and enclosed within a `$$\boxed{}$$` environment in a single
line.
- **Derivation**: Provide the subsequent mathematical derivation
that APPLIES this original formulas, assumptions, physical
principles, etc., to the problem(except for the review of the
principle itself). This derivation MUST be enclosed within an `
align` environment (e.g., `$$\begin{align} ... \end{align}$$`)
with \label{} and \tag{} for reference.
6. **Final Section**: The very last section of your output MUST
be titled `### Final Answer`. In this section, you must clearly
state the final result, and the mathematical expression for
the answer MUST be enclosed within a `$$\begin{align} ... \end{
align}$$` environment.

For example, given this problem and incomplete solution:
# Problem Statement
Find the acceleration 'a' of an apple with mass 'm', free falling
near the ground with gravitational acceleration constant 'g'.
# Solution
The gravity on the apple is F_G = mg. Using Newton's second law F
= ma, the acceleration is a = g.
Your refined solution should look like this:
# Refined Solution
### Problem Statement Explanation
This problem asks for the acceleration `a` of an apple of mass `m
`. The apple is in a state of free fall near the Earth's
surface, where the gravitational acceleration is a constant `g
`. We assume that air resistance is negligible. The goal is to
express `a` in terms of `m` and `g`.

### Step 1: Identify the Net Force on the Apple
First, we identify all forces acting on the apple. In a free-fall
scenario where air resistance is neglected, the only force is
the gravitational force exerted by the Earth. The formula for
gravitational force near the Earth's surface is:
$$
\boxed{F_g = mg}
$$
Therefore, the net force acting on the apple is equal to the
gravitational force.
$$
\begin{align}

```
F_{\text{net}} = F_g = mg
\label{eq:net_force} \tag{1}
\end{align}
$$

### Step 2: Apply Newton's Second Law of Motion
Next, we relate the net force on an object to its acceleration
    using Newton's Second Law of Motion. The standard form of the
    law is:
$$
\boxed{F_{\text{net}} = ma}
$$
We can now substitute the net force we found in Step 1 into this
    equation.
$$
\begin{align}
ma &= F_{\text{net}} \nonumber \
ma &= mg \quad (\text{using eq. \ref{eq:net_force}}) \nonumber \
a &= g
\end{align}
$$
Thus, the acceleration of the apple is equal to the gravitational
    acceleration constant `g`.

### Final Answer:
$$
\begin{align}
\boxed{a = g}
\end{align}
$$

Now I will give you the physics problem, its potentially
    incomplete solution and a report of bugs in this solution.
    Please fix ALL bugs to get a refine solution.

# Problem Statement
{question_statement}

# Solution
{solution}

# Bugs Report
{bugs_report}

 # Refined Solution
```

Subsequently, the review agent is tasked with identifying any issues in the augmented solution.

```
You are an AI expert specializing in reviewing advanced physics
    problems. Your task is to critically review the solution
    provided. You must review the solution step-by-step following
    the "### Step XXX" structure provided in the given solution.
    Carefully examine each step in the given solution. Your
    explanation of the error(s) must be concise, clear, and
    accurate.

Here is the problem and the solution to be reviewed:

# Review Instruction
When specifically reviewing each step in the given solution,
    please follow this instruction precisely:
---
{instruction}
---
After the step-by-step review, in the last line, write only '
    Correct' if every step is correct, or 'Wrong' if you find any
    errors.
```

The reviewer receives distinct instructions to evaluate the solution from multiple perspectives—such as assumptions and derivations—ensuring a comprehensive assessment.

relations / boundary conditions based on the problem statement
and the preceding (assumed correct) steps.
  – Is it a factually correct statement of an original formula /
     assumption / theorem / principle / geometric relation /
     boundary condition in its general form?
  – Is it an appropriate original formula / assumption / theorem
     / principle / geometric relation / boundary condition to
     apply at this specific stage of the problem?
.   – Is it specifically valid for this problem according to the
   problem statement?
  – Is it easily derived from more fundamental principles for
     this problem according to the problem statement, and
     therefore should not be used as an assumption?   – Is it
     specifically valid for this problem according to the problem
     statement?
3.  **Report Findings**:
  – If the original formula / assumption / theorem / principle /
     geometric relation / boundary condition is correctly
     stated and appropriately applied, state this clearly.
  – If any original formula / assumption / theorem / principle /
     geometric relation / boundary condition is misstated (e.g
     ., a sign error in the original formula) or inappropriately
     applied (e.g., using a non-relativistic formula in a
     relativistic problem) or not suitable as an assumption, you
     MUST clearly and concisely describe the error.
4.  **Conclusion**: On the final line of your response, write **
   only** `Correct` if all the original formulas / assumptions /
   theorems / principles / geometric relations / boundary
   conditions are valid and correctly applied, or `Wrong` if there
    is any error.

**Example 1 (Correct Application):**
*Step to review contains:* `\[\boxed{F_{\text{net}} = ma}\]` *for
   a dynamics problem.*
*Your review:*
The principle stated in the boxed environment is `F_net = ma`,
   which is Newton's Second Law of Motion. This is a fundamental
   and correct principle for analyzing the dynamics of a massive
   object. Its application is appropriate for this step.
Correct

**Example 2 (Incorrect Application):**
*Problem involves a block on a 30-degree incline. Step to review
   contains:* `\[\boxed{N = mg}\]` *where N is the normal force.*
*Your review:*
The principle stated is `N = mg`. While this correctly relates
   normal force to gravitational force for an object on a
   horizontal surface, it is incorrectly applied here. In this

specific problem, for a block on an inclined plane at an angle
`\theta`, the correct relation for the normal force is `N = mg
\cos\theta`. The assumption is therefore wrong for the given
geometric conditions.
Wrong

---

**Derivation Review Instruction**

In the given solution, some parts are derivations or calculations,
while others DIRECTLY adopt / introduce original formulas /
assumptions / theorems / principles / geometric relations /
boundary conditions. Your task is to judge the mathematical
derivation / calculation within the target step.

1.  **Identify the Derivation**: Focus mainly on the equations
    inside the `\[\begin{ align} ... \end{align}\]` environment.
    You should not focus on the original formulas / assumptions /
    theorems / principles / geometric relations / boundary
    conditions stated within the `\[\boxed{ }\]` environment.
2.  **Analyze Step-by-Step**: Go through the derivation line by
    line. Verify each algebraic manipulation, substitution of
    variables from previous steps (as referenced by `\label{ }` or
    `\tag{ }`), and any numerical calculations.
3.  **Report Findings**:
    - If the entire derivation is mathematically sound, state this
      .
    - If you find an error, you MUST pinpoint the exact line or
      transition where the error occurs. Quote the incorrect part
      , explain the mistake (e.g., "algebraic error," "incorrect
      substitution from tag{3}," "a sign was dropped", etc.) and
      state what the correct derivation or result should be. Use
      the `label` or `tag` for reference if available.
4.  **Conclusion**: On the final line of your response, write **
    only** `Correct` if the derivation is flawless, or `Wrong` if
    you find any mathematical error.

**Example 1 (Correct Derivation):**
*Step to review contains:* `\[\begin{align} ma &= F_{\text{net}} \
nonumber \ ma &= mg \quad (\text{using eq. \ref{eq:net_force}})
\nonumber \ a &= g \end{align}\]`
*Your review:*
The derivation begins by correctly substituting `F_net` with `mg`
based on the reference to `eq:net_force`. The final step
correctly isolates `a` by dividing both sides of `ma = mg` by `
m`. The derivation is mathematically sound.
Correct

```
**Example 2 (Incorrect Derivation):**
*Step to review contains:* `\[\begin{align} \frac{1}{2}mv_f^2 - \
    frac{1}{2}mv_i^2 &= mgh \label{eq:energy} \tag{3} \ v_f^2 - v_i
    ^2 &= \frac{1}{2}gh \end{align}\]`
*Your review:*
The derivation starts from the work-energy theorem, referenced as
    tag{3}. In the transition from the first line to the second,
    the term `m` is canceled from the left side, but the right side
     is incorrectly divided by `2m` instead of just `m`. The
    correct second line should be `v_f^2 - v_i^2 = 2gh`. This is an
     algebraic error.
Wrong
```

The outputs from these reviews are integrated into a standardized issues report template.

**Issues Report Template**

```
# Issues found in solution
## judge assumption
{issues_about_assumption}
## judge derivation
{issues_about_derivation}
```

Finally, the completed issues report is forwarded to the secretary agent for summarization, whose role is to consolidate and finalize the report.

**Summarization Prompt**

```
You are an AI expert specializing in advanced physics questions.
    Please extract, in a concise, clear and accurate manner, errors
     from a given review of a certain physics question. The errors,
     which should be extracted, have been recorded, and you MUST
    list them one by one in the format of * * Incorrect Part: * *
    and * * Explanation of Mistake: * *, in order to provide a
    clear error report.
The review text are as follows: {review}.
```

The final report can be fed back to the augmentation agent as constructive feedback, forming an iterative review loop that enables continuous refinement of the solution through multiple cycles of augmentation and critical evaluation.

In the ablation experiment, the review prompt and augmentation prompt have slight differences, as shown below.

---

**Augmentation Prompt**

```
You are a physics expert tasked with improving a solution to a
    physics problem based on review feedback.

**Problem:**
{question_statement}

**Current Solution:**
{solution}

**Review Feedback:**
{feedback}

Please provide an improved solution that addresses the issues
    mentioned in the feedback. The very last section of your output
     MUST be titled '### Final Answer'. Provide the final answer at
     the end in Latex boxed format $$\boxed{}$$. Example: $$ \boxed
    { final_answer} $$
```

---

**Reviewer Prompt**

```
You are an AI expert specializing in physics problem solving. Your
    task is to determine whether the final answer provided in a
    physics solution is correct.

You will be given:
1. A physics problem statement
2. A complete solution to that problem

Your job is to evaluate whether the final answer is mathematically
    and physically correct based on the problem requirements and
    the solution provided.

Here is the information to review:

# Problem Statement
{question_statement}

# Complete Solution
{solution}

# Instructions
```

```
Provide your analysis, then in the last line, MUST write only '
    Correct' if the final answer is correct, or 'Wrong' if the
    final answer is incorrect.
```

## A.2 EXAMPLES OF LOGICAL CHAIN AUGMENTATION

This typical example is based on question-250 from PHYbench, which is a Medium difficulty question of electromagnetic. The question and raw answer are following:

### Question

There is now an electrolyte with thickness $L$ in the $z$ direction, infinite in the $x$ direction, and infinite in the $y$ direction. The region where $y > 0$ is electrolyte 1, and the region where $y < 0$ is electrolyte 2. The conductivities of the two dielectrics are $\sigma_1, \sigma_2$, and the dielectric constants are $\varepsilon_1, \varepsilon_2$, respectively. On the $xOz$ interface of the two dielectrics, two cylindrical holes with a radius $R$ are drilled in the $z$ direction, spaced $2D$ ($D > R, R, D \ll L$) apart, with centers located on the interface as long straight cylindrical holes. Two cylindrical bodies $\pm$ are inserted into the holes, with the type of the cylinders given by the problem text below.

The cylindrical bodies $\pm$ are metal electrodes filling the entire cylinder. Initially, the system is uncharged, and at $t = 0$, a power source with an electromotive force $U$ and internal resistance $r_0$ is used to connect the electrodes. Find the relationship between the current through the power source and time, denoted as $i(t)$.

### Raw Answer

Given the potential difference $u$, it can be seen:

$$\varphi_+ = u/2, \varphi_- = -u/2, \lambda = \frac{2\pi(\varepsilon_1 + \varepsilon_2)\varphi_+}{2\xi_+} = \frac{\pi(\varepsilon_1 + \varepsilon_2)u}{\text{arccosh(D/R)}}$$

Select a surface encapsulating the cylindrical surface and examine Gauss's theorem. For the positive electrode, it is easy to see:

$$\iint \vec{E} \cdot d\vec{S} = L \oint \vec{E} \cdot \hat{n} dl = \frac{\lambda L}{(\varepsilon_1 + \varepsilon_2)/2} = \frac{2\pi u L}{2\text{arccosh}(D/R)}$$

Since the above potential distribution is deemed directly applicable for the calculation of current, the total current flowing out of the positive electrode is:

$$I = \iint \sigma \vec{E} \cdot d\vec{S} = \frac{\sigma_1 + \sigma_2}{2} \times \frac{2\pi u L}{2\text{arccosh}(D/R)}$$

Given the current $i$ passing through the power source, this current changes the net charge:

$$\frac{d(\lambda L)}{dt} = i - I = i - \frac{2\pi u L}{2\text{arccosh}(D/R)} = \frac{\pi(\varepsilon_1 + \varepsilon_2)L}{2\text{arccosh}(D/R)}\frac{du}{dt}$$

According to the loop voltage drop equation:

$$U = r_0 i + u \rightarrow u = U - r_0 i$$

$$\rightarrow i - \frac{\pi(\sigma_1 + \sigma_2)L}{2\text{arccosh}(D/R)}(U - r_0 i) = -\frac{\pi(\varepsilon_1 + \varepsilon_2)L}{2\text{arccosh}(D/R)}r_0\frac{di}{dt}$$

$$\rightarrow \frac{di}{dt} = \frac{(\sigma_1 + \sigma_2)U}{r_0(\varepsilon_1 + \varepsilon_2)} - \left(\frac{\sigma_1 + \sigma_2}{\varepsilon_1 + \varepsilon_2} + \frac{2\text{arccosh}(D/R)}{\pi r_0 L(\varepsilon_1 + \varepsilon_2)}\right)i$$

At time $t = 0$, all current should preferentially enter the capacitor. At this time, the initial current is $U/r_0$, and this differential equation yields:

$$i(t) = \frac{U}{r_0\left(1 + \frac{2\text{arccosh}(D/R)}{\pi r_0 L(\sigma_1 + \sigma_2)}\right)}\left\{2 + \frac{2\text{arccosh}(D/R)}{\pi r_0 L(\sigma_1 + \sigma_2)} - \exp\left[-\left(\frac{\sigma_1 + \sigma_2}{\varepsilon_1 + \varepsilon_2} + \frac{2\text{arccosh}(D/R)}{\pi r_0 L(\sigma_1 + \sigma_2)}\right)\right]\right\},$$

One can see missing logical steps like definitions of $u$, $\xi_+$, $\lambda$ and too brief mathematical derivations. At the first loop, LOCA's review agent gives the bug report as:

---

### Bug Report 1

Here is a report of the errors extracted from the provided review:

* **Incorrect Part:**

The foundational formula for linear charge density, $\lambda = \frac{\pi(\varepsilon_1 + \varepsilon_2)u}{\text{arccosh}(D/R)}$.

* **Explanation of Mistake:**

This formula is physically incorrect for the described system. The correct formula for capacitance per unit length, from which $\lambda$ is derived, should have $\text{arccosh}(D/(2R))$ in the denominator, not $\text{arccosh}(D/R)$. Furthermore, the correct formula has an additional factor of 2 in the denominator.

* **Incorrect Part:**

The algebraic derivation of the linear charge density $\lambda$:

$$\frac{2\pi(\varepsilon_1 + \varepsilon_2)\varphi_+}{2\xi_+} = \frac{\pi(\varepsilon_1 + \varepsilon_2)u}{\text{arccosh}(D/R)}.$$

* **Explanation of Mistake:**

There is an algebraic error in this step. Substituting $\varphi_+ = u/2$ and $\xi_+ = \text{arccosh}(D/R)$ into the middle expression yields $\frac{\pi(\varepsilon_1 + \varepsilon_2)u}{2\text{arccosh}(D/R)}$, which differs from the expression on the right by a factor of 2 in the denominator.

* **Incorrect Part:**

---

The calculation of the electric flux integral:

$$\frac{\lambda L}{(\varepsilon_1 + \varepsilon_2)/2} = \frac{2\pi u L}{2\text{arccosh}(D/R)}.$$

* **Explanation of Mistake:**

An algebraic error was made. Substituting the solution's (already incorrect) formula for $\lambda$ into the left side of the equation results in $\frac{2\pi u L}{\text{arccosh(D/R)}}$. The expression on the right side of the equality has an erroneous extra factor of 2 in the denominator.

* **Incorrect Part:**

The expression for the leakage current $I$ used in the charge conservation equation:

$$i - I = i - \frac{2\pi u L}{2\text{arccosh}(D/R)}.$$

* **Explanation of Mistake:**

This is an incorrect substitution. The expression used for $I$ is inconsistent with the one derived in the previous step

$$I = \frac{\sigma_1 + \sigma_2}{2} \times \frac{2\pi u L}{2\text{arccosh}(D/R)}.$$

It is physically incorrect as it omits the crucial effective conductivity term $(\sigma_1 + \sigma_2)/2$.

* **Incorrect Part:**

The final provided solution for the current $i(t)$.

* **Explanation of Mistake:**

The final expression for $i(t)$ is mathematically incorrect because it does not satisfy the physical conditions of the circuit. Specifically, its steady-state value (as $t \to \infty$) does not match the correct steady-state current, which should be $i_{ss} = U/(r_0 + R_{elec})$.

Due to the excessive logical leaps in the raw answer, the review agent suggests that there are multiple incorrect parts, such as definition of linear charge density $\lambda$, the algebraic derivations and final solution.

After augmentation, the logical leaps are fixed to providing a refined answer:

### Refined Answer 1

Problem Statement Explanation

The problem describes a physical system composed of two semi-infinite electrolytes, separated by the $xOz$ plane.

- **Electrolyte 1**: Occupies the region $y > 0$, with electrical conductivity $\sigma_1$ and dielectric constant (permittivity) $\varepsilon_1$.

- **Electrolyte 2**: Occupies the region $y < 0$, with conductivity $\sigma_2$ and permittivity $\varepsilon_2$.

Two long, parallel cylindrical metal electrodes, denoted as $+$ and $-$, are placed within this system.

- **Geometry**: The electrodes are of radius $R$ and length $L$. They are oriented parallel to the $z$-axis. Their centers lie on the $xOz$ interface and are separated by a distance of $2D$.

- **Assumptions**: The geometry satisfies the conditions $D > R$ and $\{R, D\} \ll L$. The latter condition allows us to neglect end effects and treat the problem as two-dimensional in the $xy$-plane, with total quantities (like capacitance and resistance) being proportional to the length $L$.

At time $t = 0$, this system is connected to a power source.

- **Power Source**: An ideal electromotive force (EMF) $U$ with a constant internal resistance $r_0$.

- **Initial Condition**: The system is initially uncharged, meaning the potential difference across the electrodes is zero at $t = 0$.

The goal is to find the current $i(t)$ flowing from the power source as a function of time.

Step 1: Equivalent Circuit Model

The physical system can be modeled as a simple electrical circuit. The two electrodes immersed in the electrolytes act as a capacitor, storing charge, and simultaneously as a resistor, allowing a leakage current to flow between them through the conductive medium.

- The capacitive nature is due to the storage of electric charge on the electrodes when a potential difference is applied, with the electrolytes acting as the dielectric material. Let's denote the equivalent capacitance as $C$.

- The resistive nature is due to the flow of charge (ions) through the electrolytes under the influence of the electric field, from the positive to the negative electrode. Let's denote the equivalent resistance of the electrolytes as $R_{elec}$.

Since charge can be stored on the electrodes while simultaneously leaking through the medium, the capacitor $C$ and the resistor $R_{elec}$ are in parallel. This parallel combination is connected in series with the internal resistance $r_0$ of the power source with EMF $U$.

Step 2: Derivation of the System's Capacitance

The total capacitance is the sum of the capacitances of the upper and lower halves of the system, which are connected in parallel.

First, we state the standard formula for the capacitance per unit length, $C'$, between two parallel cylindrical conductors of radius $R$ with centers separated by a distance $2D$ in a uniform dielectric

medium with permittivity $\varepsilon$.

$$C' = \frac{\pi\varepsilon}{\text{arccosh}(D/R)}$$

In our problem, the upper half-space ($y > 0$) is filled with electrolyte 1 ($\varepsilon_1$), and the lower half-space ($y < 0$) is filled with electrolyte 2 ($\varepsilon_2$). Due to the symmetry of the electric field about the $y = 0$ plane, we can consider this as two capacitors in parallel.

- $C_1$: Capacitance of the upper half, in a medium with permittivity $\varepsilon_1$.

- $C_2$: Capacitance of the lower half, in a medium with permittivity $\varepsilon_2$.

The capacitance per unit length for each half is:

$$C_1' = \frac{\pi\varepsilon_1}{\text{arccosh}(D/R)} \tag{1}$$

$$C_2' = \frac{\pi\varepsilon_2}{\text{arccosh}(D/R)} \tag{2}$$

The total capacitance per unit length, $C_{total}'$, is the sum of the parallel contributions.

$$\boxed{C_{total}' = C_1' + C_2'}$$

The total capacitance $C$ for the electrode length $L$ is then derived.

$$C_{total}' = \frac{\pi\varepsilon_1}{\text{arccosh}(D/R)} + \frac{\pi\varepsilon_2}{\text{arccosh}(D/R)} = \frac{\pi(\varepsilon_1 + \varepsilon_2)}{\text{arccosh}(D/R)} \tag{3}$$

$$C = C_{total}' \cdot L = \frac{\pi(\varepsilon_1 + \varepsilon_2)L}{\text{arccosh}(D/R)} \tag{4}$$

Step 3: Derivation of the System's Resistance Similarly, the total resistance of the electrolyte is determined by the two parallel paths for current flow through the upper and lower electrolytes.

For a system with a given geometry, the resistance $R$ and capacitance $C$ in a homogeneous medium are related.

$$\boxed{RC = \frac{\varepsilon}{\sigma}}$$

We can apply this relation to each half of our system. - $R_1$: Resistance of the upper electrolyte with conductivity $\sigma_1$. - $R_2$: Resistance of the lower electrolyte with conductivity $\sigma_2$.

$$R_1 = \frac{\varepsilon_1}{\sigma_1 C_1} = \frac{\varepsilon_1}{\sigma_1(C_1'L)} = \frac{\varepsilon_1}{\sigma_1 L}\frac{\text{arccosh}(D/R)}{\pi\varepsilon_1} = \frac{\text{arccosh}(D/R)}{\pi\sigma_1 L} \tag{5}$$

$$R_2 = \frac{\varepsilon_2}{\sigma_2 C_2} = \frac{\varepsilon_2}{\sigma_2(C_2'L)} = \frac{\varepsilon_2}{\sigma_2 L}\frac{\text{arccosh}(D/R)}{\pi\varepsilon_2} = \frac{\text{arccosh}(D/R)}{\pi\sigma_2 L} \tag{6}$$

Since the current can flow through both electrolytes simultaneously, these two resistances are in parallel. The total equivalent resistance of the electrolyte, $R_{elec}$, is given by:

$$\boxed{\frac{1}{R_{elec}} = \frac{1}{R_1} + \frac{1}{R_2}}$$

$$\frac{1}{R_{elec}} = \frac{\pi\sigma_1 L}{\text{arccosh}(D/R)} + \frac{\pi\sigma_2 L}{\text{arccosh}(D/R)} = \frac{\pi(\sigma_1 + \sigma_2)L}{\text{arccosh}(D/R)} \tag{7}$$

$$R_{elec} = \frac{\text{arccosh}(D/R)}{\pi(\sigma_1 + \sigma_2)L} \tag{8}$$

Step 4: Formulation of the Governing Differential Equation

Let $i(t)$ be the current from the source and $u(t)$ be the potential difference across the electrodes. Applying Kirchhoff's Voltage Law to the circuit loop:

$$\boxed{U = i(t)r_0 + u(t)}$$

The current $i(t)$ from the source splits into two paths in the parallel combination: a charging current $i_C(t)$ for the capacitor and a leakage current $i_R(t)$ through the resistor.

$$\boxed{i(t) = i_C(t) + i_R(t)}$$

The currents $i_C(t)$ and $i_R(t)$ are defined by the properties of the capacitor and resistor:

$$\boxed{i_C(t) = C\frac{du}{dt}}$$

$$\boxed{i_R(t) = \frac{u(t)}{R_{elec}}}$$

Combining these, we get a differential equation relating $i(t)$ and $u(t)$.

$$i(t) = C\frac{du}{dt} + \frac{u(t)}{R_{elec}} \tag{9}$$

To find an equation solely for $i(t)$, we eliminate $u(t)$. From the loop law, $u(t) = U - i(t)r_0$. Differentiating with respect to time gives $\frac{du}{dt} = -r_0\frac{di}{dt}$. Substituting these into Eq. equation 9:

$$i = C\left(-r_0\frac{di}{dt}\right) + \frac{U - ir_0}{R_{elec}}$$

$$i = -Cr_0\frac{di}{dt} + \frac{U}{R_{elec}} - \frac{r_0}{R_{elec}}i$$

$$Cr_0\frac{di}{dt} = \frac{U}{R_{elec}} - i\left(1 + \frac{r_0}{R_{elec}}\right)$$

$$\frac{di}{dt} = \frac{U}{Cr_0 R_{elec}} - \left(\frac{1}{Cr_0} + \frac{1}{CR_{elec}}\right)i \tag{10}$$

This is a first-order linear ordinary differential equation for $i(t)$.

Step 5: Solving the Differential Equation The differential equation equation 10 is of the form $\frac{di}{dt} + Bi = A$.

$$\boxed{\frac{dy}{dt} + P(t)y = Q(t) \implies y(t) = e^{-\int P(t)dt}\left(\int Q(t)e^{\int P(t)dt}dt + K\right)}$$

For our constant-coefficient case, the general solution is $i(t) = i_{ss} + i_h(t) = A/B + Ke^{-Bt}$, where $i_{ss}$ is the steady-state current and $K$ is a constant determined by the initial condition.

$$B = \frac{1}{Cr_0} + \frac{1}{CR_{elec}} = \frac{r_0 + R_{elec}}{Cr_0 R_{elec}} \tag{11}$$

$$i_{ss} = \frac{A}{B} = \frac{U/(Cr_0 R_{elec})}{(r_0 + R_{elec})/(Cr_0 R_{elec})} = \frac{U}{r_0 + R_{elec}} \tag{12}$$

The initial condition at $t = 0$ is that the system is uncharged, so $u(0) = 0$. From the loop law, $U = i(0)r_0 + u(0)$, which gives the initial current:

$$i(0) = \frac{U}{r_0} \tag{13}$$

Applying this to the general solution at $t = 0$:

$$i(0) = i_{ss} + Ke^0$$

$$\frac{U}{r_0} = \frac{U}{r_0 + R_{elec}} + K$$

$$K = \frac{U}{r_0} - \frac{U}{r_0 + R_{elec}} = U\frac{(r_0 + R_{elec}) - r_0}{r_0(r_0 + R_{elec})} = \frac{UR_{elec}}{r_0(r_0 + R_{elec})} \tag{14}$$

Substituting $i_{ss}$, $K$, and $B$ back into the solution gives $i(t)$:

$$i(t) = \frac{U}{r_0 + R_{elec}} + \frac{UR_{elec}}{r_0(r_0 + R_{elec})} \exp\left(-\frac{r_0 + R_{elec}}{Cr_0 R_{elec}}t\right) \tag{15}$$

Final Answer

The relationship between the current through the power source and time, $i(t)$, is found by substituting the expressions for the equivalent capacitance $C$ (Eq. equation 4) and resistance $R_{elec}$ (Eq. equation 8) into the general solution (Eq. equation 15).

The steady-state current is:

$$i_{ss} = \frac{U}{r_0 + R_{elec}} = \frac{U}{r_0 + \frac{\text{arccosh}(D/R)}{\pi(\sigma_1 + \sigma_2)L}}$$

The decay constant in the exponent is:

$$B = \frac{1}{CR_{elec}} + \frac{1}{Cr_0} = \frac{\sigma_1 + \sigma_2}{\varepsilon_1 + \varepsilon_2} + \frac{\text{arccosh}(D/R)}{\pi r_0(\varepsilon_1 + \varepsilon_2)L}$$

The final expression for the current $i(t)$ is:

$$i(t) = \frac{U}{r_0 + \frac{\text{arccosh}(D/R)}{\pi(\sigma_1 + \sigma_2)L}} + \left(\frac{U}{r_0} - \frac{U}{r_0 + \frac{\text{arccosh}(D/R)}{\pi(\sigma_1 + \sigma_2)L}}\right) \exp\left[-\left(\frac{\sigma_1 + \sigma_2}{\varepsilon_1 + \varepsilon_2} + \frac{\text{arccosh}(D/R)}{\pi r_0(\varepsilon_1 + \varepsilon_2)L}\right)t\right] \tag{8}$$

This answer is obviously more detailed and easier to verify. The review agent accurately locate the error:

---

**Bug Report 2**

Here is the error report based on the provided review:

* * Incorrect Part: * *

Step 2: Derivation of the System's Capacitance.

* * Explanation of Mistake: * *

The solution incorrectly applies the formula for capacitance per unit length between two full parallel cylinders,

$$C' = \frac{\pi \varepsilon}{\operatorname{arccosh}(D/R)},$$

to a system involving two half-spaces. The correct capacitance for each half-space is only half of this value. By omitting the factor of 1/2, the solution calculates a total capacitance 'C' that is twice the correct value.

---

Referring to this bug report, LLM generated the correct answer with correct steps in the next iteration. This answer received recognition from both the reviewer agent and human expert. The correct answer is

---

**Correct Answer**

$$i(t) = \frac{U}{r_0 + \frac{2\operatorname{arccosh}(D/R)}{\pi(\sigma_1+\sigma_2)L}}$$

$$+ \left( \frac{U}{r_0} - \frac{U}{r_0 + \frac{2\operatorname{arccosh}(D/R)}{\pi(\sigma_1+\sigma_2)L}} \right) \exp\left[ -\left( \frac{\sigma_1+\sigma_2}{\varepsilon_1+\varepsilon_2} + \frac{2\operatorname{arccosh}(D/R)}{\pi r_0(\varepsilon_1+\varepsilon_2)L} \right) t \right],$$

---

which is different from the raw answer. This example demonstrates that Logical Chain Augmentation not only improves the detection of errors within detailed solution steps, but also enables the LLM to generate the correct reasoning path and final answer.

### A.3 THE USE OF LLMS

LLMs were used to help polish writing, improve conciseness, and check grammar across the sections of the paper.

