# OpenReview forum: "LOCA: Logical Chain Augmentation for Scientific Corpus Cleaning"
_ICLR.cc/2026/Conference — Submitted to ICLR 2026_

### Official Review · Reviewer_dSxn · 2025-10-24

**Soundness:** 2
**Presentation:** 3
**Contribution:** 2
**Rating:** 4
**Confidence:** 4

**Summary:**

LOCA proposes an agent-based workflow for quality filtering of scientific QA data (in practice only physics QA is used). The method consists of three main components:
1. Logic-chain enhancement: Complete the implicit reasoning steps in the original answer and decompose each step into a pair: (Principle, Derivation).
2. Iterative review: Two reviewers (agents) independently verify the principle and derivation of each step. A QA pair is accepted if it passes three consecutive review rounds; conversely, if it accumulates five failures, it is discarded.
3. Final check: The enhanced final answer must remain identical to the original answer.
Experiments demonstrate a significant reduction in residual error rate on multiple physics QA datasets.

**Strengths:**

- The idea of augmenting hidden reasoning chains is novel and interesting.
- On three physics QA datasets the method shows strong empirical improvement.
- The (Principle, Derivation) pairing naturally matches reasoning patterns in scientific problems and provides a clear structure for human auditing.

**Weaknesses:**

- Although the title claims to target scientific corpora, the evaluation is limited to physics, leaving generalization to other domains unverified.
- Using a single LLM for both enhancement and reviewing may introduce self-bias and may partially explain the advantage observed under evaluation with Gemini 2.5 Pro.
- The contribution is more of a engineering design rather than algorithmic innovation.

**Questions:**

- Consider validating whether the cleaned dataset further improves downstream model training quality.
- Consider evaluating variants that incorporate multiple distinct models into the enhancement/review loop to avoid single-model bias.

---

### Official Review · Reviewer_ro15 · 2025-10-30

**Soundness:** 2
**Presentation:** 2
**Contribution:** 2
**Rating:** 2
**Confidence:** 4

**Summary:**

This paper presents LOCA (Logical Chain Augmentation), a framework for automatically cleaning scientific corpora by addressing logical incompleteness in reasoning chains. The method employs an augment-and-review loop that (1) completes missing logical steps through chain completion, (2) decomposes each step into orthogonal principle and derivation components, and (3) uses specialized review agents to iteratively refine solutions. Applied to physics QA datasets (PHYBench, PHYSICS, ABench-Physics), LOCA reportedly reduces error rates from ~20% to below 2% while maintaining substantial dataset size.

**Strengths:**

1. The paper addresses a genuine problem—high error rates in scientific reasoning datasets—with a structured approach. The principle-derivation decomposition provides interpretable intermediate representations that could facilitate both automated review and human verification.

2. The evaluation compares against diverse baselines spanning reasoning methods (CoT, ToT, GoT), review methods (Review-SC), and iterative refinement (Self-Reflection) across multiple LLMs. The detailed example in Appendix A.2 illustrates how the method processes a physics problem through multiple iterations.

**Weaknesses:**

1. Ground truth is created by experts using LOCA's structured outputs to identify errors, then LOCA is evaluated against these same errors. This makes the <2% error rate claim unverifiable.

2. Only 300 questions evaluated across three datasets, far too limited for a method claiming to enable "large-scale, high-quality scientific corpora." No computational cost analysis provided despite requiring up to 8 LLM calls per question.

3. Please provide Cohen's kappa across multiple independent experts for error identification to validate ground truth reliability.

4. Bottom margins appear excessively wide throughout, resulting in significantly less content per page than standard ICLR submissions. Authors must verify compliance with official ICLR 2026 template.

**Questions:**

1. What are computational costs? Report: (a) average LLM calls per question, (b) total tokens processed, (c) wall-clock time, (d) cost per 1000 questions, (e) comparison with baselines.

2. What are hyperparameter sensitivities? Show how error rate and accepted set size vary across N_{corr}^{(max}} ∈ {1,2,3,4,5} and N_{wrg}^{(max)} ∈ {1,3,5,7,10}.

3. How does the system determine when a step is "non-atomic"? How are principles identified from axiom space P? How is semantic equivalence determined for external consistency checks?

4. Can you confirm template compliance? The bottom margins appear non-standard. Verify the manuscript uses the official ICLR 2026 LaTeX template without modifications.

---

### Official Review · Reviewer_Vcy6 · 2025-11-01

**Soundness:** 2
**Presentation:** 2
**Contribution:** 2
**Rating:** 2
**Confidence:** 3

**Summary:**

The paper introduces LOCA, a framework for automatically cleaning scientific QA corpora through logical chain augmentation. LOCA reconstructs reasoning by inserting missing logical steps and separating each step into principles and derivations. Experiments on three physics benchmarks show that LOCA reduces corpus error rates from around 20% to below 2%.

**Strengths:**

LOCA identifies an  issue (logical incompleteness of solution) in problem-solving benchmarks and proposes a solution to tackle this.

Authors show empirical results across PHYBench, PHYSICS, and ABench-Physics. It shows that LOCA can reduce the residual error rate compared to baselines.

**Weaknesses:**

- The framework mainly applies to problem-solving questions in derivation-heavy domains like physics or mathematics, while the title and introduction claim a broader scope across scientific domains. This seems inaccurate since corpora in other domains can contain other issues such as factual errors, unclear writing, or formatting problems. Clarifying the scope would make the paper more accurate.
- The definition of “error” appears to focus on logical incompleteness rather than final-answer correctness. In my opinion, it is somewhat unclear whether a solution with a correct final answer but missing intermediate steps should be considered erroneous. The paper would benefit from clarifying what kinds of reasoning flaws LOCA is designed to detect, and why such augmentation matters if a knowledgeable reader could easily fill in those steps.
- Evaluation benchmarks are limited to physics. Including results from other scientific fields, such as math, would strengthen the generalizability of LOCA.
- The iterative review loop depends on LLM judgment. However the paper lacks analysis of the reliability of these LLM-based reviewers. The paper would benefit from some human evaluation of review quality.
- The dataset used for experiments is relatively small: only 100 questions per benchmark are sampled from larger datasets (e.g., PHYBench with 500 problems, PHYSICS with 1297).
- Hyperparameter choices (Ncorr, Nwrg) are not justified or analyzed for sensitivity.

**Questions:**

1. How sensitive are results to the number of review iterations (Ncorr, Nwrg)? How they chosen in the paper?
2. In lines 38–39, the authors state that “our own expert analysis reveals that error rates in major benchmarks’ QA pairs can exceed 20%.” How is this “error” defined? What criteria or guidelines were provided to the experts? Does it include both incorrect answers (as shown in the appendix) and logically incomplete solutions?

---

### Official Review · Reviewer_V6rY · 2025-11-03

**Soundness:** 1
**Presentation:** 1
**Contribution:** 1
**Rating:** 2
**Confidence:** 3

**Summary:**

(I am relatively confused by the contribution of this paper, but I will try my best to summarize the work given my understanding)

This paper proposes an agentic pipeline, LOCA, that can clean language model responses for solving some physics problems: given a generated reasoning paragraph, LOCA breaks it down or rewrites the original generation into atomic reasoning units, conducts logical reasoning, and then rejects examples that fail to satisfy the logical constraints. The authors conduct some evaluation of the proposed agent pipeline, and claim that it can be used to clean datasets (without any experimental results supporting that).

**Strengths:**

Unfortunately I cannot say I am excited about specific aspects of this paper, since I am very confused about the paper’s presentation (see my comments in the weakness section below).

**Weaknesses:**

I think my confusion comes from the mismatch between the experiments as well as the goal/claim of this paper.
1. If I understand correctly, the authors aim to develop a pipeline to clean existing (training) corpus for scientific use cases. Therefore, one way to validate this idea seems to be (1) build a method (2) validate the correctness of the cleaning method and (3) apply the method to clean some corpus, and show the usefulness (in terms of the improved accuracy when applying the trained models). However, the paper only has (1) and (2), but not (3). The author only uses this agentic framework to clean answers for a small-scale evaluation dataset, and uses the residual error rate to demonstrate its effectiveness. There’s nothing about (3) in the paper.
2. And even for (2), I don’t think the current evaluation is sufficient: It uses the residual error rate and retention corpus size as the evaluation metrics. The authors mentioned that they’ve already used a LOCA filtered subset (line 278) for this experiment – as such, there’s significant bias in the reported numbers.

**Questions:**

How is “residual error rate” defined? Does it mean the error rate in the filter examples? Please define it properly in the paper.

---

### Meta-Review · Area_Chair_SBc9 · 2025-12-29

**Summary:**

This work proposes LOCA, a pipeline designed to clean scientific corpora by augmenting logical chains and using an iterative review loop. While the problem of low-quality scientific data is significant, the committee has decided to reject this work due to fundamental flaws in the evaluation methodology and scope. The authors did not submit a rebuttal, leaving substantial reviewer concerns unaddressed.

**Reviewer Concerns:**

The authors did not provide a rebuttal and engage in the discussion phase. Hence, the raised concerns by reviewers are still unsolved. Reviewer V6rY identified a critical gap in the experimental design, noting that the paper builds a cleaning pipeline but fails to validate if the cleaned data actually improves downstream model performance. Reviewer Vcy6 criticized the discrepancy between the broad scientific claims in the title and experiments limited solely to physics, alongside ambiguous definitions of error. Reviewer ro15 raised a serious methodological concern regarding the circularity of the evaluation, where the ground truth appears to be generated by the tool itself rather than independent verification. Reviewer dSxn, while acknowledging the engineering effort, flagged the risk of single-model bias and the lack of generalization beyond physics derivation problems.

**Reviewer Scores:**

Based on the above issues, I think all the reviewers would keep their scores.

---

### Decision · Program_Chairs · 2026-01-26

Reject